# Numerical Study of the Interaction between Level Ice and Wind Turbine Tower for Estimation of Ice Crushing Loads on Structure

**Ming Song [1], Wei Shi [2], Zhengru Ren [3] and Li Zhou [1,*]**

[1] School of Naval Architecture and Ocean Engineering, Jiangsu University of Science and Technology, Zhenjiang 212003, China; songmingcssc@163.com

[2] State Key Laboratory of Coast and Offshore Engineering, Deepwater Engineering Research Center, Dalian University of Technology, Dalian 116024, China; weishi@dlut.edu.cn

[3] Department of Marine Technology, Norwegian University of Science and Technology, Trondheim No-7451, Norway; zhengru.ren@ntnu.no

[*] Correspondence: zhouli209@hotmail.com

**Abstract:** In this paper, the interaction between level ice and wind turbine tower is simulated by the explicit nonlinear code LS-DYNA. The isotropic elasto-plastic material model is used for the level ice, in which ice crushing failure is considered. The effects of ice mesh size and ice failure strain on ice forces are investigated. The results indicate that these parameters have a significant effect on the ice crushing loads. To validate and benchmark the numerical simulations, experimental data on level ice-wind turbine tower interactions are used. First, the failure strains of the ice models with different mesh sizes are calibrated using the measured maximum ice force from one test. Next, the calibrated ice models with different mesh sizes are applied for other tests, and the simulated results are compared to corresponding model test data. The effects of the impact speed and the size of wind turbine tower on the comparison between the simulated and measured results are studied. The comparison results show that the numerical simulations can capture the trend of the ice loads with the impact speed and the size of wind turbine tower. When a mesh size of ice model is 1.5 times the ice thickness, the simulations can give more accurate estimations in terms of maximum ice loads for all tests, i.e., good agreement between the simulated and measured results is achieved.

**Keywords:** ice crushing; ice load; finite element method; wind turbine tower; numerical simulation

## Nomenclature

| | |
|---|---|
| OWT | Offshore Wind Turbine |
| HAWC2 | Horizontal Axis Wind turbine simulation Code 2nd generation |
| FAST | Fatigue Aerodynamics Structures and Turbulence |
| Std. | Standard Deviation |
| $\varepsilon^p$ | Effective plastic strain |
| t | Time |
| $D_{ij}^p$ | Plastic component of the rate of deformation tensor |
| $F_f$ | Ice force in full scale |
| $F_m$ | Ice force in model scale |
| $\lambda$ | Experimental scale |
| $F_{mean}$ | Mean force |
| $F_i$ | Ice force at each time |
| N | Total number of output force |
| $\sigma$ | Standard deviation of force |

## 1. Introduction

With the growing renewable energy demands and the increasing concern about environmental pollution, the development of renewable energy harnessing has been paid more and more attention. As a sort of clean energy, wind energy has become the most promising renewable energy after decades of development. Compared with the land-based wind, offshore environment has more abundant wind energy resources with higher quality, and OWTs could avoid the problems of land acquisition and noise [1]. However, a key technology challenge for OWTs is operation in cold climates, i.e., the possibility of the structure interaction with floe ice enhances while operating in cold regions. For example, the Great Lakes are the most promising locations for the OWTs in the United States. The lakes are often substantially ice covered for the entire winter, and have wind and sea current driven ice floes at times [2]. An OWT operating under wind and ice conditions is shown in Figure 1. Ice loads should be taken as one of the important environmental impacts in addition to the aerodynamic loads. Therefore, it is necessary to predict the ice loads caused by the level ice-OWT interaction.

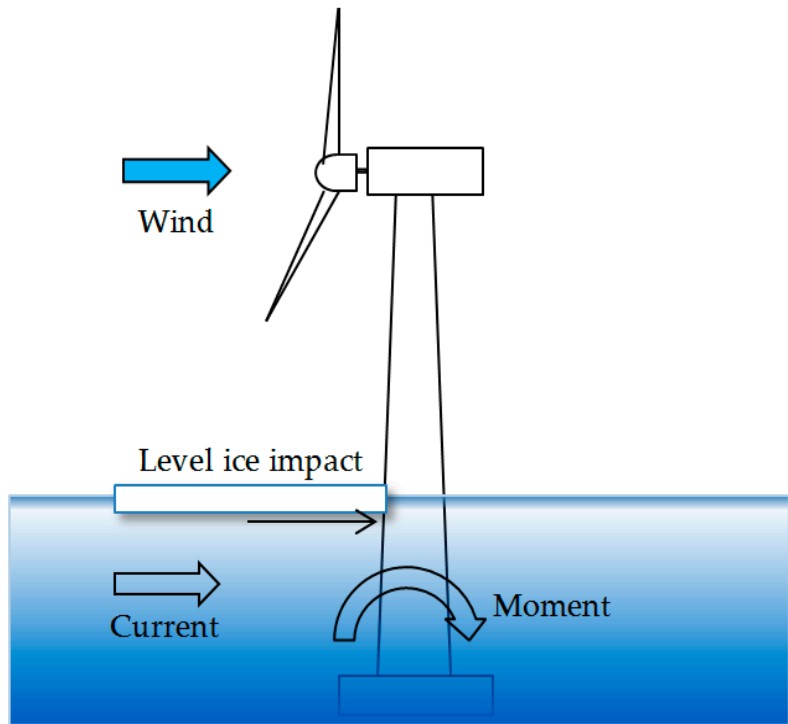

**Figure 1.** OWT exposed to wind field and in contact with ice.

The interaction between an OWT and level ice is a complex process. The magnitude and time variation of the ice loads depend strongly on the geometry of the wind turbine, the ice thickness, the ice strength, and their impact velocity. There are various ice failure modes observed in the ice-structure interaction. Typically, for level ice, bending, buckling, cracking/splitting, or crushing could take place, which is strongly governed by the shape of the structure at the water level [3,4]. The sloping shapes cause the level ice to fail by bending, whereas the vertical shapes induce the level ice to fail by crushing [5]. The level ice is weaker in bending than crushing. Therefore, the implementation of sloping shapes for OWTs can effectively reduce the magnitude of ice loads. Many model tests have verified the lower ice loads on a conical structure than on a cylindrical structure of similar size [6–8]. However, the application of sloping shapes will induce larger wave loads and enhance the foundation costs because of the additional material near the water level.

Many studies have been carried out to investigate the interaction between ice and cylindrical or sloped structures by model tests, full-scale tests and numerical simulations [9–14]. Yue et al.

conducted full-scale tests on a cylindrical compliant monopod platform to investigate the dynamic ice forces and structure vibrations generated by crushing failure of the ice sheet [15]. The test results showed that three ice force modes take place in the loading speeds which make ice fail in ductile, ductile-brittle transition, and brittle range respectively. Kuutti et al. simulated ice crushing against a rigid vertical structure using cohesive surface methodology [16]. The simulated crushing forces agreed well with the experimental results. Lu et al. and Wang et al. carried out numerical simulations of interactions between level ice and sloping structure using the cohesive element method [17,18]. Zhou et al. proposed a numerical model to simulate the non-simultaneous crushing force acting on the cylindrical structures of wind turbines [19]. It was observed that the simulation results agree well with the measured data from the model tests in terms of the maximum ice force. Ranta et al. simulated ice rubble-structure interaction processes based on arbitrary Lagrangian-Eulerian finite element method [20]. However, there was a lack of validation on the characteristics of the obtained rubble pile geometries.

Some researchers focus on the predictions of the coupled dynamic loads and responses of an OWT [2,21]. Shi et al. studied the dynamic ice-structure interaction of a monopile-type OWT in drifting level ice in both parked and operating conditions by coupling a semi-empirical numerical model to the aero-hydro-servo-elastic simulation tool HAWC2 [22]. The effects of ice drifting speed and ice thickness were investigated by using the coupled dynamic analyses. It was found that the effect of the ice thickness on the response is significant, whereas the effect of drifting speed on the bending moment response in the fore-aft direction is negligible. Wells developed a simulation tool to study the effects of ice on both cylinder- and cone-shaped OWTs [23]. The simulation results indicated that the surface ice sheet loads can be much larger than the wind loads and could be the driving parameters of the OWT foundations design in areas where ice can occur. Heinonen and Rissanen conducted a feasibility study of the FAST simulation software to investigate the structural performance of OWTs [4]. They studied the ice interaction with vertically shaped structures at the water line and taking into account the coupling between the ice, wind, and structural response. However, there is a limitation in the ice model for describing a brittle crushing process.

For the load design, the ice crushing is the most important since it causes the biggest force and might induce severs steady-state vibrations as well [24]. When the cylindrical structures are interacting with drifting ice (of thickness 0.2 m and more), the ice crushing failure action can generate as high dynamic forces as 5 MN and are of critical concern for the structural designers [25]. Therefore, it is necessary to investigate the dynamic interaction between level ice and vertical structures where ice crushing failure takes place. Most of the present works established the ice forces from the existing ice models, in which the dynamic interaction process and the ice crushing failure could not be simulated.

This paper focuses on the numerical study to predict the ice crushing force acting on the cylindrical OWT foundation based on the nonlinear finite element method, in which the dynamic interaction process is simulated. The isotropic elasto-plastic material model is used for level ice to simulate ice crushing failure. The effects of ice mesh size and failure strain on the ice forces are investigated. Model tests on the interaction between level ice and nearly vertical wind turbine tower are used to calibrate and validate the numerical simulation results. Four impact cases are considered. The comparisons between the simulated and measured results including the maximum, mean, standard deviation, and time series of the ice forces are made. In addition, the studies on the effects of the impact speed and the size of wind turbine tower on the comparison are carried out.

## 2. Experimental Data

This section presents the experimental data used to calibrate and verify the numerical simulation results. The tests were conducted by Wu et al. at [26] the ice Basin of Tianjin University. The interaction between level ice and wind turbine tower was considered. The experimental scale was chosen to be 1:20. The force in full scale $F_f$ is calculated by the following equation:

$$F_f = F_m \lambda^3 \tag{1}$$

The test represents the impacts between a 0.4 m thickness level ice and the monopile foundations of a 3-MW and a 4-MW wind turbine towers at speeds varying from 0.05 to 1.2 m/s in full scale. The target thickness of the level ice is 0.4 m in full scale. The bending and crushing strength of the level ice are expected to be 0.6 MPa and 2.06 MPa in full scale, respectively. A total of 12 impact tests were conducted. Tests #304, #306, #404, and #406 are selected for the analysis because the brittle ice crushing failure took place in these tests and the time histories of ice forces for these tests were available. The specific test matrix and the ice properties are given in Table 1.

**Table 1.** Test matrix and measured ice conditions (in full scale).

| Test | Wind Turbine Tower | Ice Thickness (m) | Bending Strength (kPa) | Crushing Strength (kPa) | Ice Drifting Speed (m/s) |
|------|------|------|------|------|------|
| #304 | 3 MW | 0.4 | 572 | 1980 | 0.6 |
| #306 | 3 MW | 0.4 | 664 | 2122 | 1.2 |
| #404 | 4 MW | 0.4 | 572 | 1980 | 0.6 |
| #406 | 4 MW | 0.4 | 664 | 2122 | 1.2 |

Figure 2 shows the geometry of the 3-MW and 4-MW monopile wind turbine towers. The foundations of the wind turbine towers are nearly vertical structures. The diameter of the 3-MW and 4-MW monopile wind turbine towers at waterline is 5.30 m and 5.83 m in full scale, respectively, and their slope angle is 87.2 degrees and 88.3 degrees, respectively.

An ice force experiment scenario for a 3 MW model test is shown in Figure 3. A force transducer measured the ice loads using a data acquisition system with a sampling frequency of 100 Hz.

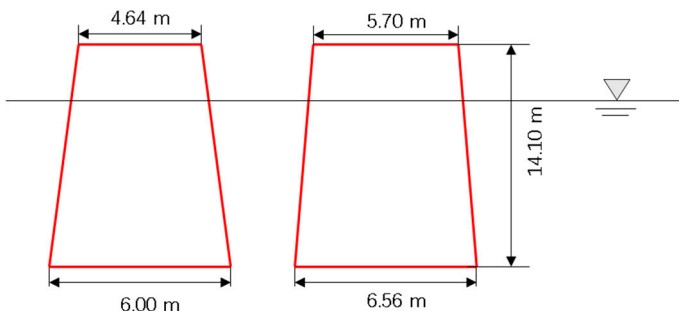

**Figure 2.** The geometry of the 3-MW and 4-MW wind turbine towers in full scale.

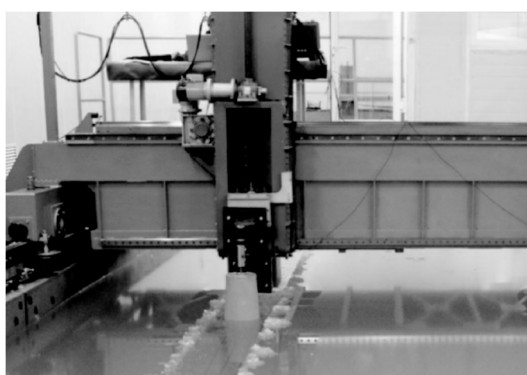

**Figure 3.** Photograph of a 3-MW model test from Zhou et al. [19].

## 3. Numerical Analysis

This section details the finite element modeling, the material models, and major results. All simulations were run on an 8 CPU workstation with Inter 3.60 GHz processors and 16.0 GB of RAM. The software used was LS-DYNA version R700 with double precision. LS-DYNA software has a number of contact algorithms and a large suite of material types that can be chosen for the interacting structures. It has been widely used to simulate ice-structure collisions. Patran software was used for the modeling and generation of meshes for the study.

### 3.1. Model Description

Figure 4 shows the numerical domain of the simulations. The dimensions of the level ice are 55 m × 55 m × 0.4 m. The ice model extent is sufficient to minimize the effect of boundary conditions. The dimensions of the wind turbine towers in the numerical models are the same as the experimental models.

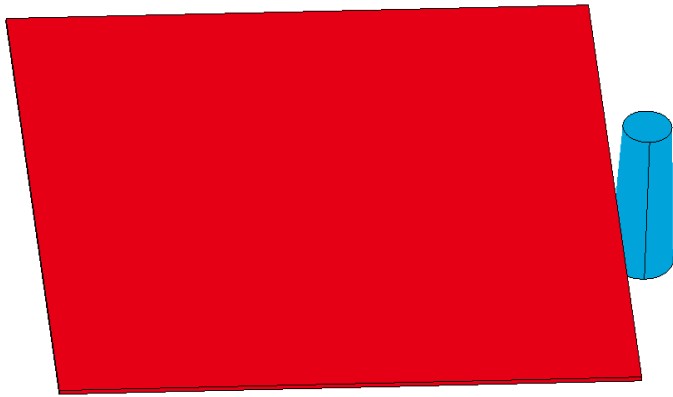

**Figure 4.** Numerical model of ice-structure interaction.

The ice is modeled using eight-node solid elements. The wind turbine towers are discretized using four-node Belyscho-Tsay shell elements. The mesh size for the level ice is approximately 0.6 m × 0.6 m × 0.4 m, in which there is only one layer of meshes in the vertical direction. The wind turbine towers are meshed with an element size of 0.4 m. The number of elements for the ice and the wind turbine towers are 151250 and 1645, respectively.

To avoid the initial penetration and numerical instabilities, the translational velocity of the wind turbine tower at water plane ramps up from 0.0 m/s to 1.2 m/s before the impact occurs. After it reaches 1.2 m/s, the velocity is kept to be constant throughout the rest of the simulations. This can be achieved by using the LS-DYNA command "boundary prescribed motion rigid" with define curve. LS-DYNA offers a large number of contact types. The contact between the level ice and the wind turbine tower is implemented through the contact-eroding-surface-to-surface formulation, which is used with the segment-based contact option (soft = 2) in LS-DYNA. This eroding contact type contains logic which allows the contact surface to be updated to consider the ice element deletion [27]. The ice is defined as "slave "segment and the wind turbine tower is defined as "master" segment, a search for penetration of a "slave" node through the "master" segment is made every time step. When a penetration is found, a contact force proportional to the penetration depth is applied to resist and ultimately eliminate the penetration. The contact force is contained in the "rcforc" file produced by using a database-rcforc command. In order to consider the self-contact of the ice component, the contact-eroding-single-surface contact type which is the most widely used contact options in LS-DYNA is applied for the ice model. Both static and dynamic coefficients of friction are set to 0.15 at all the contacts, which is a reasonable assumption for the friction between the ice and the steel surfaces.

### 3.2. Material Models

For finite element analyses of ice-structure interactions, the constitutive material model for the ice is a critical factor to accurately predict maximum ice forces [28]. Wang et al. proposed an ice model for the interaction between sloping marine structure and level ice by using the cohesive element model [18]. In their model, the isotropic elasto-plastic linear softening constitutive model proposed by Hilding et al. was introduced to present the microscopic crushing of the ice sheet, while the bending failure of ice sheet was caused by the failure of cohesive elements [29]. In our case, the slope angle of the wind turbine towers is close to 90 degree. The ice crushing is in the dominant failure mode during the interaction between the level ice and the wind turbine towers. Therefore, the isotropic elasto-plastic material model is used for the level ice in this paper.

Figure 5 shows the relationship between the yield stress and effective plastic strain for the ice model. The effective plastic strain is defined as:

$$\varepsilon^p = \int_0^t (\frac{2}{3} D_{ij}^p D_{ij}^p)^{1/2} dt \tag{2}$$

The ice performance is assumed to have three stage states: The ice material is elastic before reaching the crushing initial point; after the first crack, the ice material shows a linear softening behavior; when the ice is totally crushed, it behaves as a viscous fluid. To describe the ice behavior, the "mat-piecewise-linear-plasticity" material type from LS-DYNA's suite of material types is used here, in which an elasto-plastic material with the yield stress versus strain curve and failure based on a plastic strain can be defined. For the wind turbine towers, the rigid material model is used, in which the deformation of the structure during interaction is not considered. The input material parameters to both the level ice and the wind turbine towers models are given in Table 2.

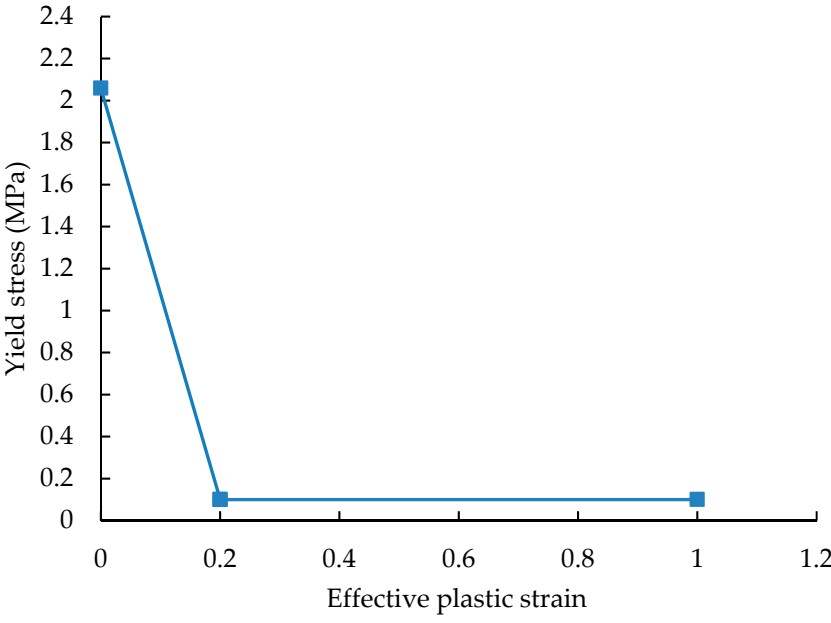

**Figure 5.** Hardening curve for ice material model.

**Table 2.** Material parameters used in the simulations.

| Items | Level Ice | Wind Turbine Towers |
|---|---|---|
| Density (kg/m³) | 900 | 7850 |
| Young's modulus (GPa) | 2 | 210 |
| Poisson ratio (-) | 0.3 | 0.3 |
| Yield stress (MPa) | 2.06 | - |
| Failure strain (-) | 0.2 | - |

### 3.3. Effect of Ice Mesh Size

To investigate the effect of the ice mesh size on the ice force, four meshes with characteristic element lengths of 0.2 m, 0.4 m, 0.6 m, and 0.8 m are considered, and corresponding size ratio (mesh size/ ice thickness) is 0.5, 1, 1.5, and 2, respectively. It is noted that there are two layers of meshes in the vertical direction for mesh size of 0.2 m, and only one layer for other mesh size (shown in Figure 6). The ice failure strain of 0.2 is used in all simulations. The other parameters are equal to the basic values according to the setup of test #306.



**Figure 6.** Side view of ice model: (**a**) mesh size of 0.2 m (**b**) mesh size of 0.8 m

Figure 7 shows the horizontal ice force histories for different ice mesh sizes. It is found that the ice mesh size has a significant effect on both the fluctuated frequency and peak forces. With the refinement of mesh, the frequency increases, while the peak forces decrease. The mean and standard deviation of force are calculated by the following equations:

$$F_{mean} = \frac{1}{N} \sum_{i}^{N} F_i \tag{3}$$

$$\sigma = \sqrt{\frac{1}{N} \sum_{i}^{N} (F_i - F_{mean})^2} \tag{4}$$

The comparison of the mean, standard deviation, and maximum forces are tabulated in Table 3. The simulated maximum force varies from 1.88 MN to 4.34 MN. Figure 8 shows the mean, standard deviation, and maximum forces varying with the mesh size. It is shown that the simulation with coarse mesh yields higher standard deviation and maximum force. Overall, both the standard deviation and maximum forces present an approximately linear relationship with the mesh size. However, for the mean force, the simulated values with different mesh sizes are around 0.86. The simulated results indicate that the mesh size has a significant effect on both the standard deviation and the maximum loads, while it has a slight effect on the mean load.

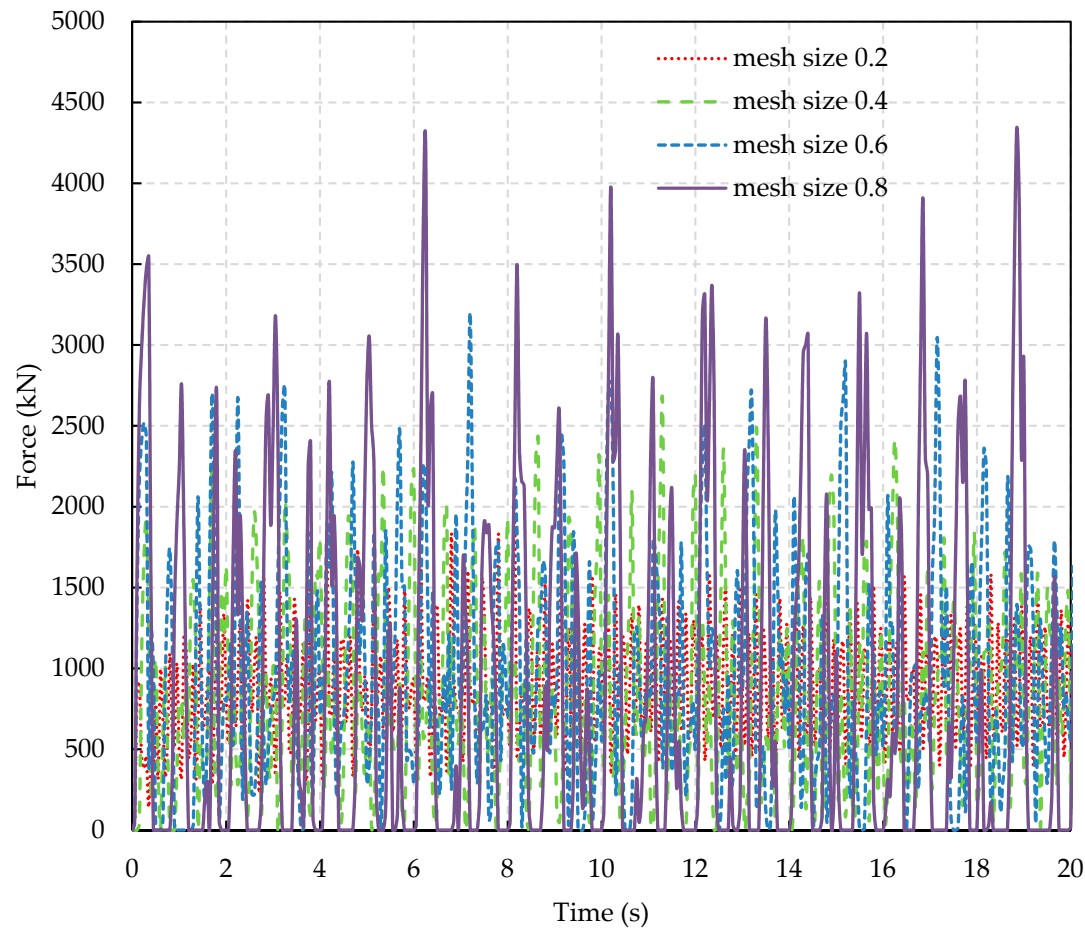

**Figure 7.** Ice force histories from the simulations with different mesh sizes.

**Table 3.** Comparison of the ice forces for the simulations with different mesh sizes.

| Mesh Size (m) | 0.20 | 0.40 | 0.60 | 0.80 |
|---|---|---|---|---|
| Mean (MN) | 0.88 | 0.83 | 0.98 | 0.87 |
| Std. (MN) | 0.35 | 0.59 | 0.70 | 1.11 |
| Maximum (MN) | 1.88 | 2.65 | 3.20 | 4.34 |

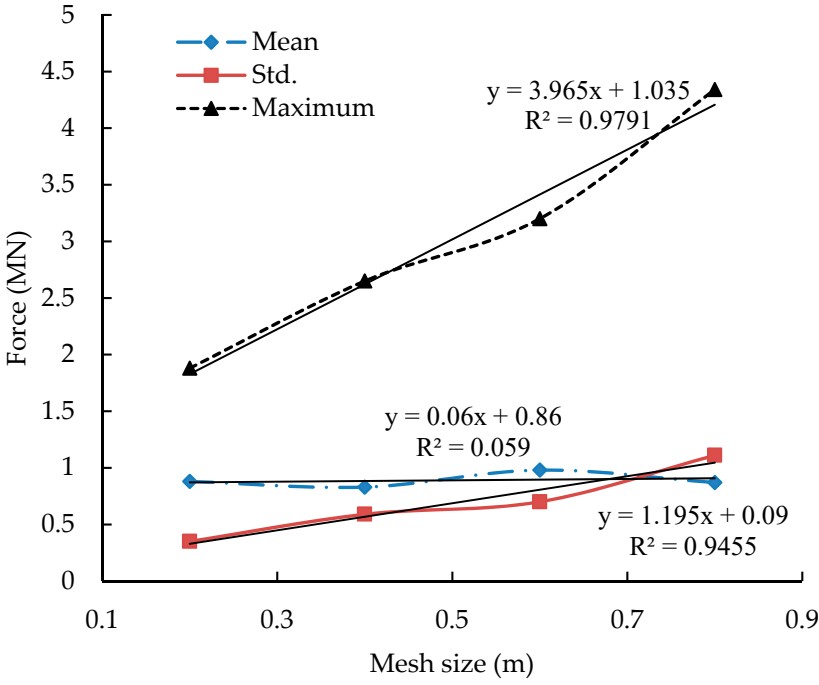

**Figure 8.** The mean, std. and maximum forces varying with the mesh size.

### 3.4. Effect of Ice Failure Strain

To investigate the effect from the ice failure strain, numerical simulations of the interaction between the level ice and the 3 MW wind turbine tower with different failure strain coefficients are carried out. The values of the ice failure strain in different simulations are set as 0.15, 0.2, 0.25, and 0.3, respectively. The other parameters are kept constant and equal to the basic values according to the setup of test #306 where the drift speed is 1.2 m/s in full scale.

Figure 9 shows the comparison of the horizontal ice force histories for various failure strains. It is observed that the fluctuated frequencies in the four curves are similar. The peak load increases with increasing failure strain. The mean, standard deviation, and maximum values are listed in Table 4 and these values varying with failure strain is shown in Figure 10. It is seen that the mean, standard deviation, and maximum forces present a linear increasing tendency with the larger failure strain. The linear curves which are fitted to the simulated mean, standard deviation, and maximum data are also presented in the figure, in which the slope of the curves is 4.48, 1.76, and 11.68, respectively. The simulated results indicate that the failure strain plays an important role in the simulated ice forces.

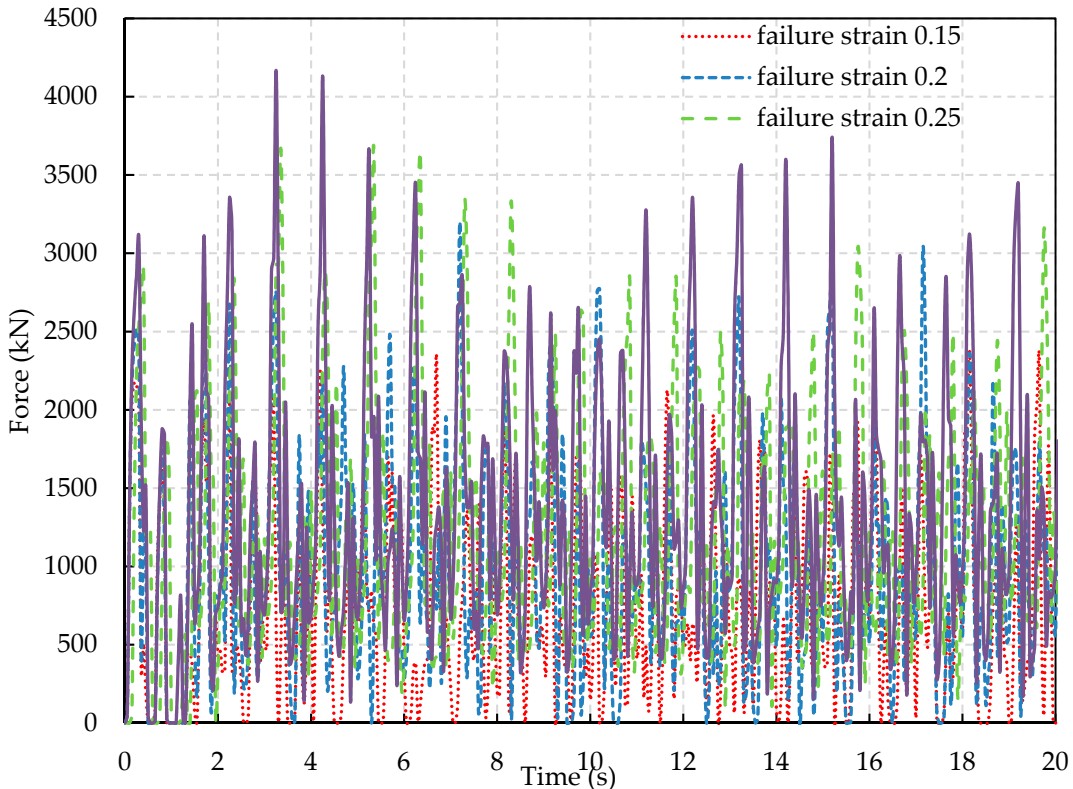

**Figure 9.** Ice force histories from the simulations with different failure strains.

**Table 4.** Comparison of the ice forces for the simulations with different failure strains.

| Failure Strain | 0.15 | 0.20 | 0.25 | 0.30 |
|---|---|---|---|---|
| Mean (MN) | 0.70 | 0.98 | 1.21 | 1.37 |
| Std. (MN) | 0.59 | 0.70 | 0.77 | 0.86 |
| Maximum (MN) | 2.38 | 3.20 | 3.67 | 4.17 |

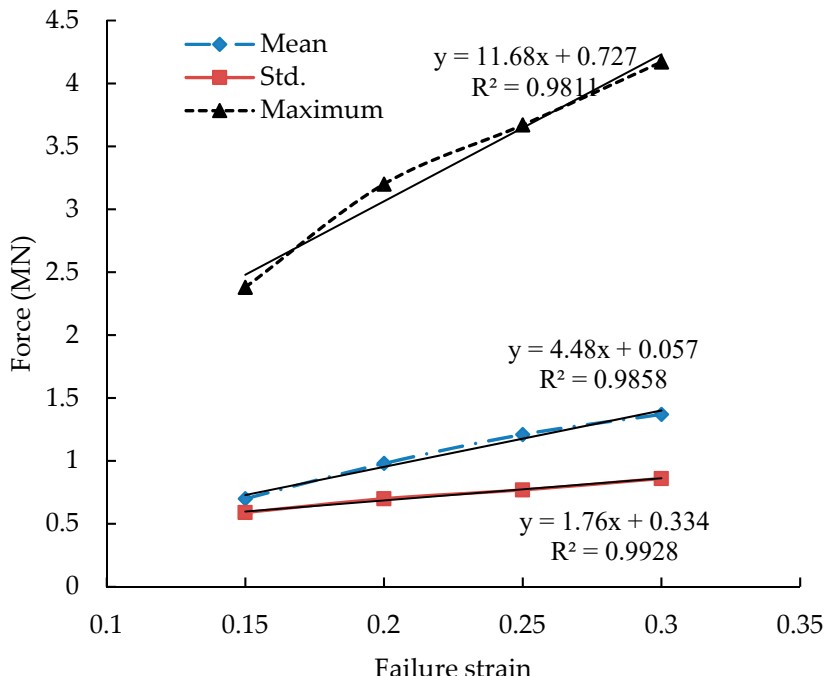

**Figure 10.** The mean, standard deviation, and maximum forces varying with the failure strain.

In summary, both the ice failure strain and the ice mesh size are crucial to ice force, including the mean, standard deviation, and maximum values. In addition, the fluctuated frequency is lower in the simulation with coarse mesh. Therefore, the failure strain should be determined from the numerical simulation with a given mesh size and the parameters should be calibrated using available experimental data.

## 4. Comparison of the Numerical Simulations and Test Results

This section presents the comparisons of the horizontal ice force histories, maximum, mean, and standard deviation values between the simulated and measured results for tests #306, #304, #404, and #406. Four groups of meshes are considered.

### 4.1. Comparison of Test #306

According to the results from section 3.3 and 3.4, the selections of ice failure strain for different ice mesh sizes are justified by a trial and error procedure which yields the better results for the maximum load, i.e., the simulated maximum force for the interaction between the level ice and the 3-MW wind turbine tower is in good agreement with the experimental measurement for test #306.

The ice failure strains of 0.43, 0.29, 0.2, and 0.12 are determined for using in the numerical simulations with the mesh size of 0.2 m, 0.4 m, 0.6 m, and 0.8 m, respectively. The relationship between the failure strain and the size ratio is shown in Figure 11. It is observed that the failure strain decreases with increasing size ratio. In this figure, an exponential curve $y = Ce^{Ax}$, where $C = 0.67$ and $A = -0.84$, is fitted to the simulation data. It is shown that the difference between the two curves is small, in which the coefficient of determination $R^2$ is equal to 0.99. The results indicate that $y = 0.67e^{-0.84x}$ can be adopted to describe the relationship between failure strain and size ratio for this case.

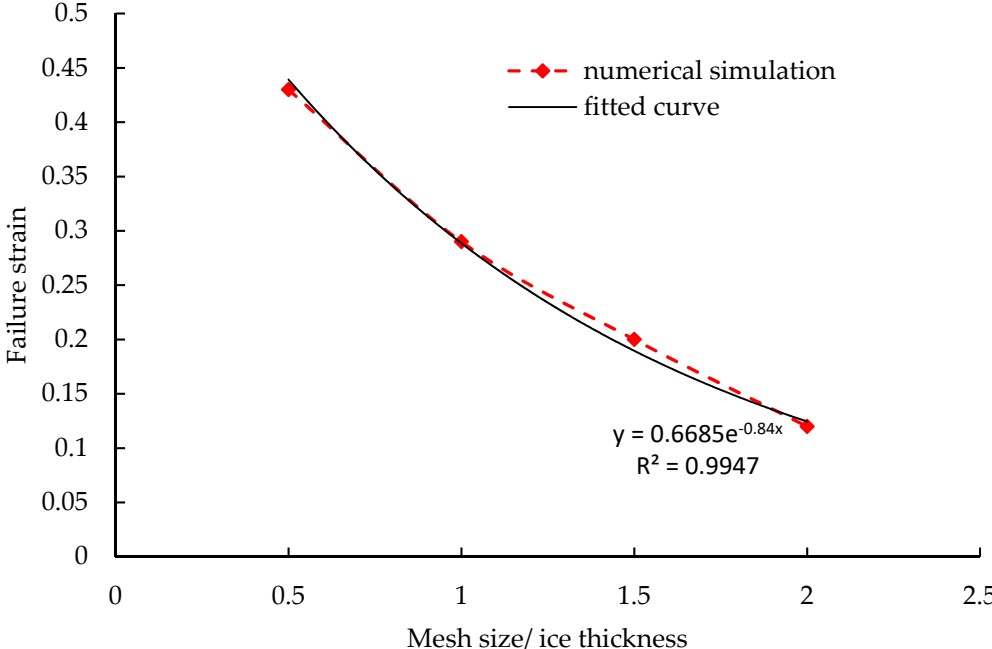

**Figure 11.** Relationship between failure strain and size ratio.

Figures 12–15 show the comparison of horizontal ice force histories between the simulations and measurement for test #306. In general, both the simulated and measured ice forces present strong nonlinear and vibrating behavior, and their trends are similar. It is observed that all simulations

capture the maximum force well. Overall, the simulation with mesh size of 0.2 m gives better results: most of the peak and valley values are around 2.8 MN and 0.8 MN, respectively (see the dash line in Figure 12), which are in good agreement with the model test. However, the valley values in the other simulations are much smaller than the measurement, especially in the simulation with mesh size of 0.8 m. It can be seen that the zero forces obtained from the simulation with mesh size of 0.8 m are much more than those obtained from the other simulations and the model test (see Figure 15). This is mainly because the accumulation and sliding forces from the interaction between the wind turbine tower and the ice fragments are not considered in the simulations where the failed ice elements are removed. With increasing mesh size of ice, the simulated ice breaking length increases, and consequently the gap between the wind turbine tower and the unbroken ice sheet will increase.

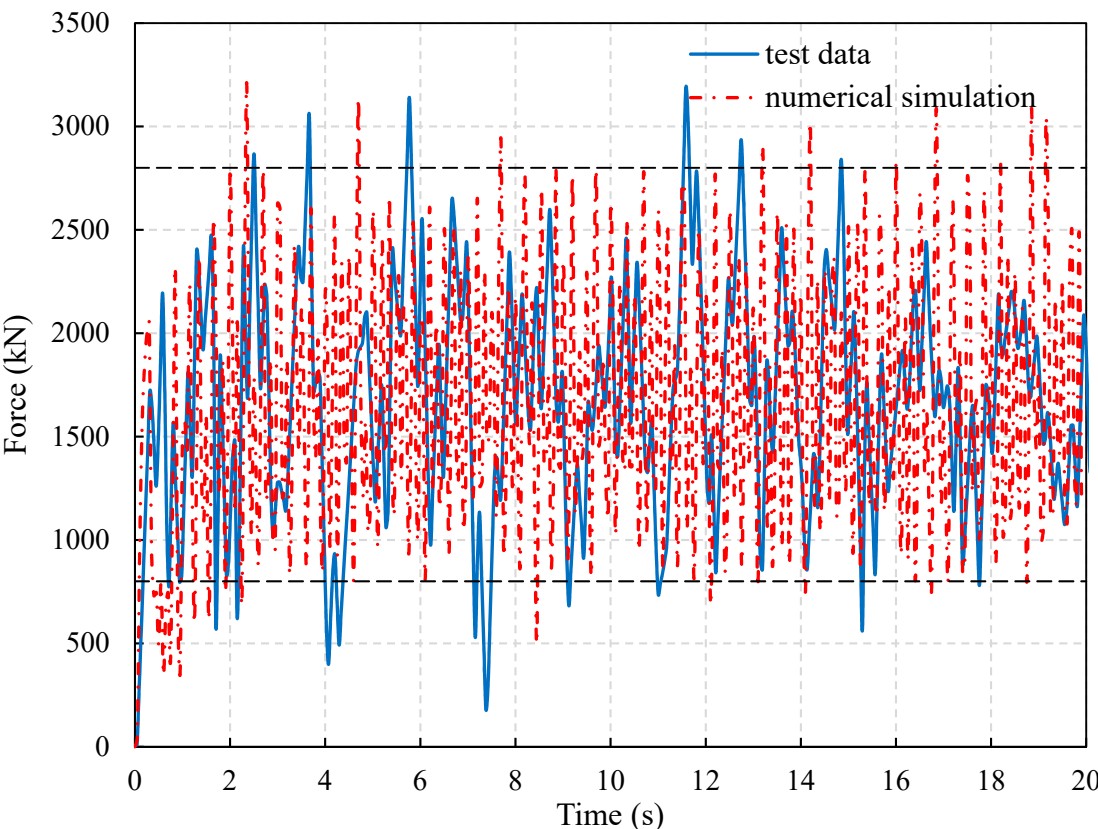

**Figure 12.** Ice force histories from the simulation with mesh size of 0.2 m and measurement for test #306.

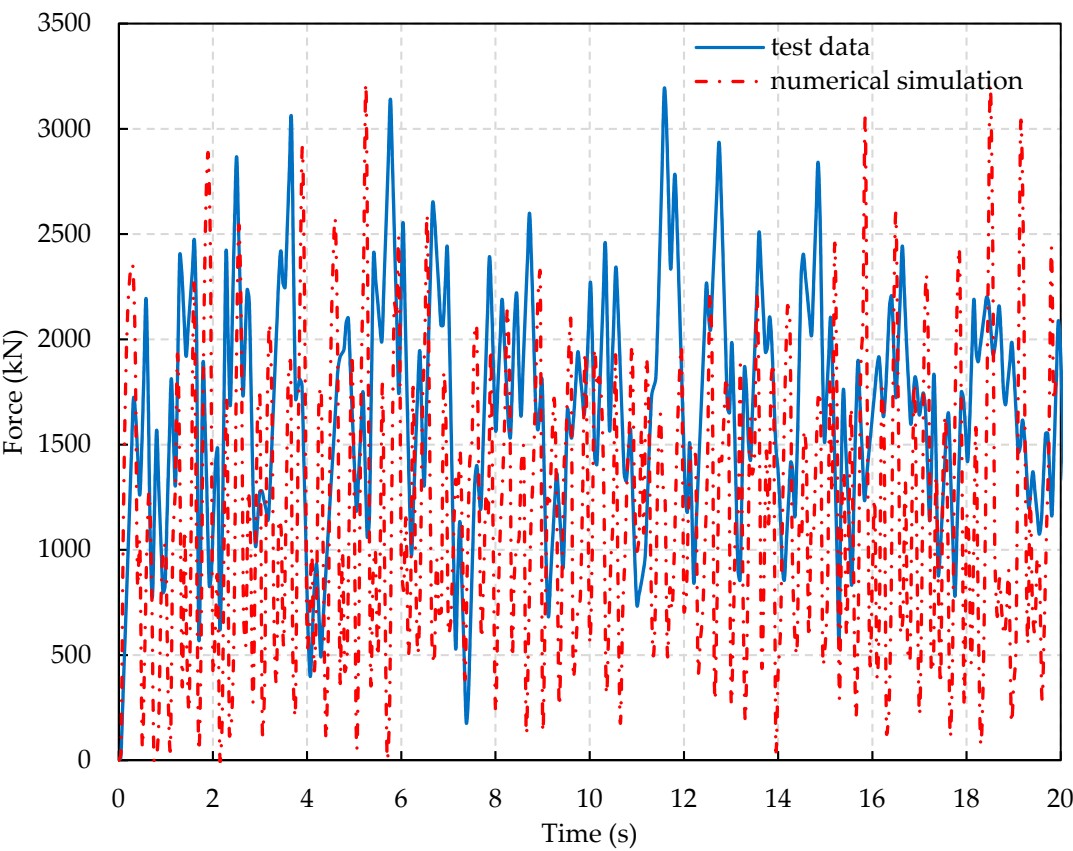

**Figure 13.** Ice force histories from the simulation with mesh size of 0.4 m and measurement for test #306.

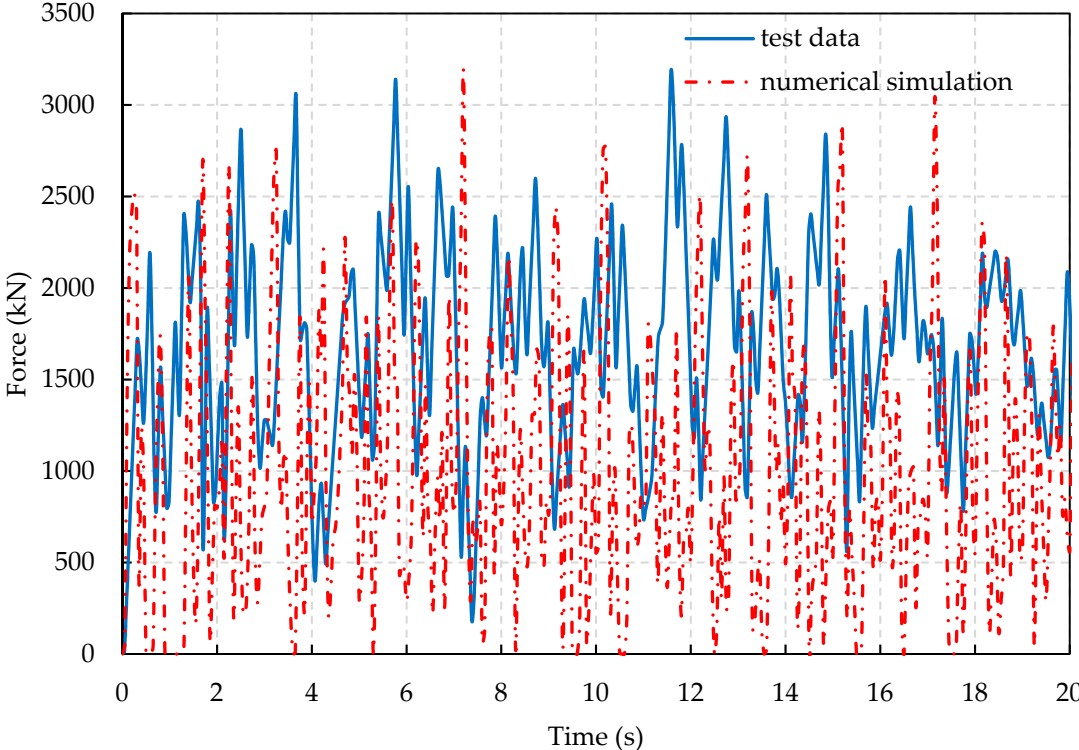

**Figure 14.** Ice force histories from the simulation with mesh size of 0.6 m and measurement for test #306.

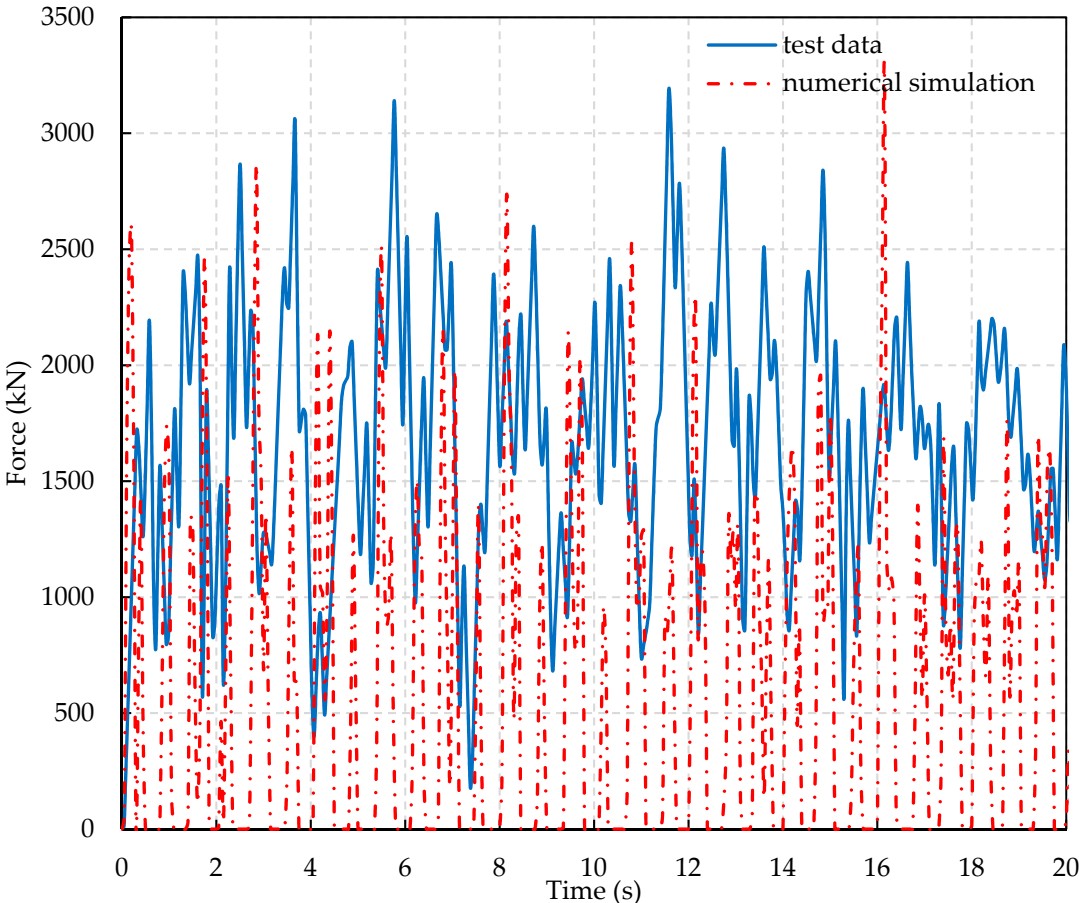

**Figure 15.** Ice force histories from the simulation with mesh size of 0.8 m and measurement for test #306.

To have a better understanding of the icebreaking process, Figures 16–17 give the partial magnification snapshots of the ice sheet at t = 10 s in the simulation with mesh sizes of 0.2 m and 0.8 m. It is obviously shown that there is a big gap between the wind turbine tower and the unbroken ice sheet in the simulation with mesh size of 0.8 m, which results in many zero forces. For the simulation with mesh size of 0.2 m, full contact between the structure and the ice can be found, which means that the interaction between the structure and the level ice is continuous. This phenomenon can explain why the mesh size affects the forces. Figure 18 shows an ice crushing scenario from test #306. It is seen that there are many small ice pieces accumulating in front of the wind turbine tower during the interaction. However, there is no interaction between the wind turbine tower and the broken ice pieces as failed ice elements are removed in the simulations. This limitation of element deletion in the numerical simulations results in the differences of the ice forces between the simulated and measured results.

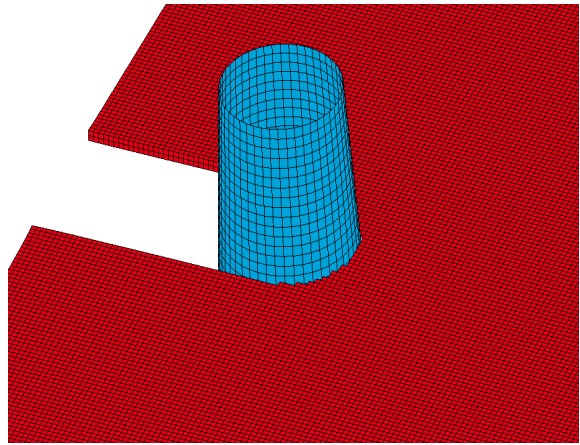

**Figure 16.** An image extracted from the simulation with mesh size of 0.2 m at t = 10 s.

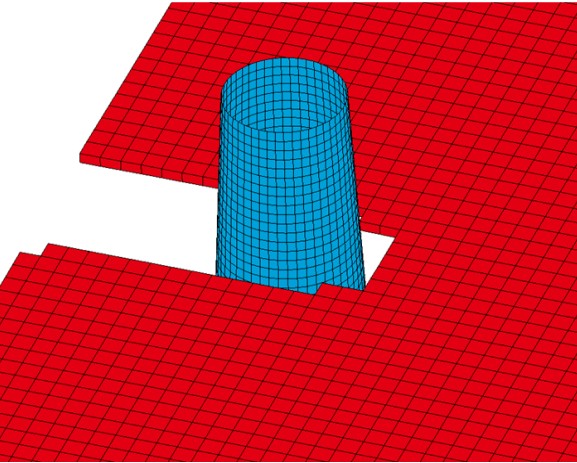

**Figure 17.** An image extracted from the simulation with mesh size of 0.8 m at t = 10 s.

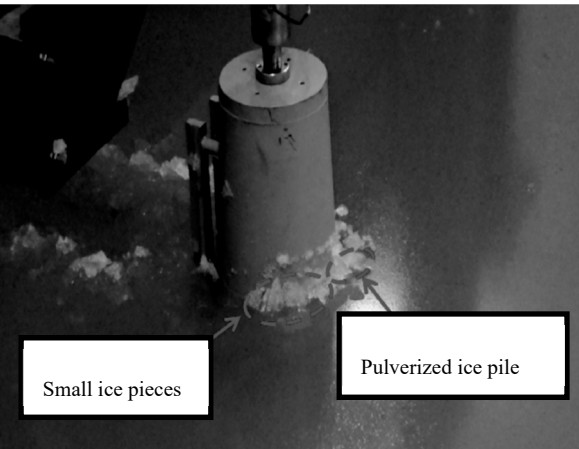

**Figure 18.** An image extracted from test #306.

The simulated results including the mean, standard deviation, and maximum loads for different mesh sizes are presented in Table 5. In addition, the corresponding model test data are given for comparison. The comparison shows that the discrepancy of the maximum ice force between the simulated and measured results is small for all mesh sizes. The largest discrepancy is 2.82% for mesh size of 0.8 m, and the smallest is only 0.3% for mesh size of 0.6 m. However, the mean load decreases with increasing mesh size. It is also found that the simulation with mesh size of 0.2 m provides better predictions of mean load and standard deviation than the other simulations, i.e., there is a good

agreement between the simulation with mesh size of 0.2 m and the experiment. The discrepancies of the mean load and the standard deviation are 3.1% and 3.3%, respectively. The simulations when the mesh size is larger than 0.2 m underestimate the mean load, while overestimating the standard deviation. This is due to more zero forces in the simulations with larger mesh size.

**Table 5.** Comparison between the simulated and measured results for test #306.

| Items | Numerical Simulations (MN) | | | | Model Test (MN) |
|---|---|---|---|---|---|
| | Mesh size 0.2 m | Mesh size 0.4 m | Mesh size 0.6 m | Mesh size 0.8 m | |
| Maximum | 3.21 | 3.21 | 3.20 | 3.28 | 3.19 |
| Mean | 1.64 | 1.16 | 0.98 | 0.55 | 1.59 |
| Std. | 0.62 | 0.67 | 0.70 | 0.70 | 0.60 |

Figures 19–22 show the spectra of the ice force from the measurement and simulations with different mesh sizes for test #306. It is observed that the main frequency in the measurement is 1.05 Hz, compared to 6.05 Hz in the simulation with mesh size of 0.2 m, 3.05 Hz in the simulation with mesh size of 0.4 m, 2.05 Hz in the simulation with mesh size of 0.6 m, and 1.55 Hz in the simulation with mesh size of 0.8 m, which indicates that the main frequency in the simulation with larger mesh size is lower and closer to the experimental data. It is noted that there is a good agreement of a peak between the simulation with mesh size of 0.6 m and the measurement at the frequency at 1.05 Hz.

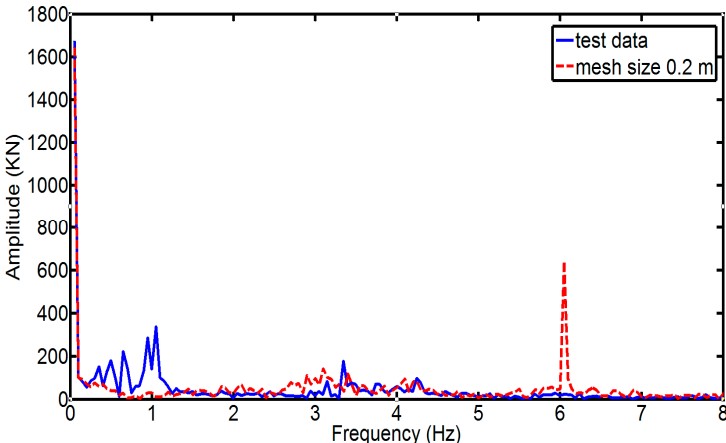

**Figure 19.** Spectrum of ice force from the simulation with mesh size of 0.2 m and measurement for test #306.

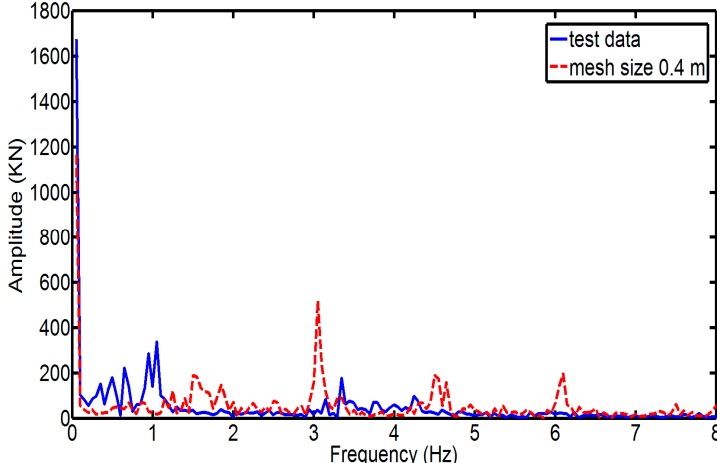

**Figure 20.** Spectrum of ice force from the simulation with mesh size of 0.4 m and measurement for test #306.

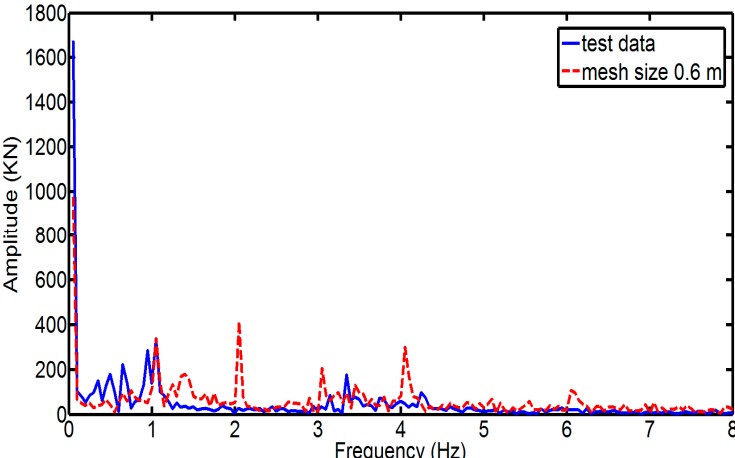

**Figure 21.** Spectrum of ice force from the simulation with mesh size of 0.6 m and measurement for test #306.

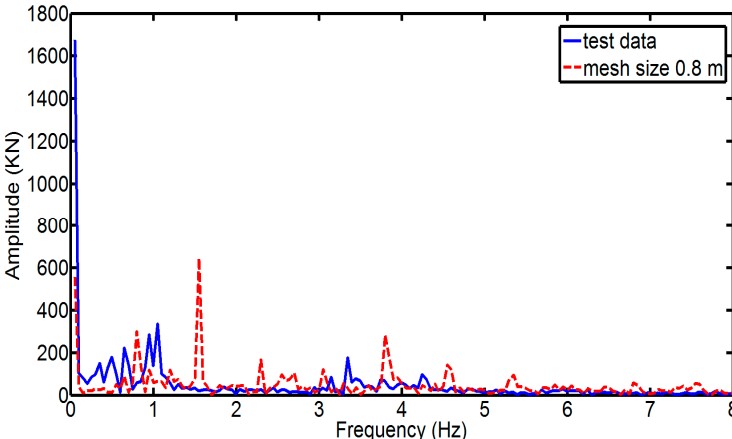

**Figure 22.** Spectrum of ice force from the simulation with mesh size of 0.8 m and measurement for test #306.

In summary, the simulations when the relationship between failure strain and size ratio is $y = 0.67e^{-0.84x}$ can provide an accurate prediction of maximum ice force for test #306. When comparing the ice force histories between the simulated and measured results, more zero forces are found in the simulations when the mesh size is larger than 0.2 m. Besides, the simulations overestimate the main frequency than the model test.

### 4.2. Comparison of Test #304

It is interesting to investigate the effect of the impact speed on the comparison between the simulated and measured results. Therefore, test #304 is simulated, in which the impact speed is 0.6 m/s. The other parameters and mesh size are the same as they are in Section 4.1.

The time series of the horizontal ice forces from both the simulations and the measurements for test #304 are given in Figures A1–A4. The comparisons show similar ice force characteristics with those for test #306. The maximum, mean, and standard deviation values derived from the simulated and measured ice forces are presented in Table 6. The comparison shows that the simulation with mesh size of 0.6 m provides a more accurate prediction of maximum ice force than the other simulations. The discrepancy between all simulated and measured maximum force ranges from 0.6%

to 6.9%. In addition, the mean ice force decreases with increasing mesh size, and the mean value obtained from the simulation with mesh size of 0.4 m is closest to the measured data. All simulations overestimate the standard deviation.

It is concluded that there is a good agreement between all simulated and measured results with regard to the maximum ice force, which is consistent with the conclusion for test #306. The results indicate that the impact speed has little effect on the comparison of maximum ice forces between the simulations with different mesh sizes and the model tests.

**Table 6.** Comparison between the simulated and measured results for test #304.

| Items | Numerical Simulations (MN) | | | | Model Test (MN) |
|---|---|---|---|---|---|
| | Mesh size 0.2 m | Mesh size 0.4 m | Mesh size 0.6 m | Mesh size 0.8 m | |
| Maximum | 3.25 | 3.28 | 3.47 | 3.37 | 3.49 |
| Mean | 1.63 | 1.13 | 0.94 | 0.57 | 1.15 |
| Std. | 0.66 | 0.72 | 0.75 | 0.73 | 0.60 |

*4.3. Comparison of the Tests #404 and 406*

To investigate the effects of the dimension of the wind turbine tower on the comparison between the simulated and measured results, test #404 and test #406 are also simulated, in which a 4 MW wind turbine tower is used. The drift speeds in test #404 and test #406 are 0.6 m/s and 1.2 m/s, respectively. The other parameters and mesh size are the same as they are in Section 4.1.

The maximum, mean, and standard deviation values obtained from both the simulated and measured ice forces for test #404 are presented in Table 7. It is observed that the maximum ice force calculated from the simulation with mesh size of 0.6 m fits well with the measured result in which the discrepancy is 2.1%. The discrepancy between the other simulations and the measured results ranges from 10.8% to 21.2%. For the mean force, the simulation with mesh size of 0.2 m is closest to the measured results. In addition, the standard deviation is higher than the model test data when the mesh size is larger than 0.2 m.

The comparisons for test #406 are shown in Table 8. Similar results with test #404 are found. The maximum ice force obtained from the simulation with mesh size of 0.6 m and the mean force obtained from the simulation with mesh size of 0.2 m are closest to the measured results, in which the discrepancies are 9.3% and 1.8%, respectively. The discrepancy of maximum force between the other simulations and the measured results ranges from 13.3% to 23.7%.

**Table 7.** Comparison between the simulated and measured results for test #404.

| Items | Numerical Simulations (MN) | | | | Model Test (MN) |
|---|---|---|---|---|---|
| | Mesh size 0.2 m | Mesh size 0.4 m | Mesh size 0.6 m | Mesh size 0.8 m | |
| Maximum | 3.35 | 4.71 | 4.16 | 4.85 | 4.25 |
| Mean | 1.73 | 1.35 | 0.85 | 0.75 | 1.58 |
| Std. | 0.49 | 1.01 | 1.02 | 1.00 | 0.85 |

**Table 8.** Comparison between the simulated and measured results for test #406.

| Items | Numerical Simulations (MN) | | | | Model Test (MN) |
|---|---|---|---|---|---|
| | Mesh size 0.2 m | Mesh size 0.4 m | Mesh size 0.6 m | Mesh size 0.8 m | |
| Maximum | 3.26 | 4.61 | 4.11 | 4.65 | 3.76 |
| Mean | 1.74 | 1.37 | 0.84 | 0.76 | 1.71 |
| Std. | 0.45 | 0.97 | 0.96 | 0.93 | 0.72 |

## 5. Discussion and Conclusions

In this paper, numerical simulations of the interaction between level ice and wind turbine tower have been performed using software LS-DYNA. The study confirms that both the mesh size and the failure strain of the ice model play a significant role in the simulated ice forces. With the refinement of mesh size, the simulated maximum ice force decreases, while the fluctuated frequency of ice force increases. The mesh size influences both the maximum load and the load frequency greatly. This finding is similar to those in Wang et al. and Lu et al. for simulating ice-sloping structure interactions [17,18]. It is also found that the mean, standard deviation, and maximum values derived from the simulated ice forces increase with increasing failure strain. The failure strain of the ice model is not a strictly material property but rather a numerical remedy to excessive mesh distortions. Therefore, its application to the simulation of a physical phenomenon requires the calibration with experimental results.

In our study, the measured maximum ice force derived from test #306 is used to calibrate the failure strain of the ice model with different mesh sizes. The relationship between failure strain and size ratio is obtained, i.e., $y = 0.67e^{-0.84x}$. It is found that a larger failure strain should be applied for the simulations with smaller mesh size to achieve an accurate prediction of maximum ice force. This is due to the combined effect of mesh size and failure strain on the simulated ice force. In order to investigate the effect of the impact speed and the wind turbine tower diameter, the same numerical models are applied to simulate the interaction for tests #304, #404, and #406.

It is found that a mesh size (0.6 m) that is 1.5 times the ice thickness (0.4 m) predicts maximum ice force with reasonable accuracy for all tests (see Figure B1), in which the discrepancy between the simulations and the model tests ranges from 0.3% to 9.3%. The size ratio (mesh size/ice thickness) is similar to that in Wang et al. for simulating the interaction between sloping marine structure and the level ice [18].

The other simulations can provide accurate predictions of maximum force for test #304 and #306 (see Figure B1). It indicates that the impact speed has little effect on the comparison between the numerical simulations with different mesh sizes and the model tests. These numerical models can be used to study the effect of speed for the same impact objects with regard to maximum force.

The discrepancy of maximum ice force between the simulated and measured results for test #404 and #406 ranges from 2.1% to 23.7%, which is larger than that for test #304 and #306 (i.e., 0.3% to 6.9%). It indicates that the predictive accuracy may decrease when the impacted structure is changed.

It is also found that the simulated maximum forces for tests #404 and 406 are higher than those for tests #304 and #306 (see Figure B1). This is because the diameter of the wind turbine tower at the mean waterline in tests #404 and #406 are relatively larger. Besides, when comparing the simulated maximum ice forces for tests #304 and #306, or tests #404 and #406, it is seen that slower impact speed results in a larger maximum force. These are confirmed by the model test results.

There exists significantly discrepancy of mean load and standard deviation between the simulated and measured results (see Figures B2–B3). This is mainly caused by the limitations of the ice model. As an element deletion technique is used to remove failed ice elements from the calculation, numerical simulation of ice crushing generates zero contact that is created upon the deletion of elements. In addition, the rotation, accumulation and sliding forces that are contributed by the crushed ice could not be considered using this ice model. These limitations of the ice model will be addressed in the future work.

It should be noted that the model tests with the experimental scale of 1:20 were used for the comparison with the numerical simulations. The effect of the experimental scale on the comparison will be investigated by using more physical tests including model and full scale tests in the future work.

**Author Contributions:** Conceptualization, M.S.; methodology, M.S.; software, M.S.; validation, M.S., L.Z., and W.S.; formal analysis, M.S.; investigation, M.S.; resources, W.S. and Z.R.; data curation, W.S.; writing—original draft preparation, M.S.; writing—review and editing, W.S., Z.R.; visualization, M.S.; supervision, L.Z.; project administration, L.Z.; funding acquisition, L.Z., W.S.

**Funding:** This research was funded by the National Natural Science Foundation of China, grant number 51809124, 51911530156, 51709039; Natural Science Foundation of Jiangsu Province of China, grant number BK20170576; Natural Science Foundation of the Higher Education Institutions of Jiangsu Province of China, grant number 17KJB580006; State Key Laboratory of Ocean Engineering (Shanghai Jiao Tong University), grant number 1704 and 1807.

**Acknowledgments:** The authors would like to thank the Jiangsu University of Science and Technology (JUST).

**Conflicts of Interest:** The authors declare no conflict of interest.

## Appendix A. The time series of the horizontal ice forces from both the simulations and measurement for test #304

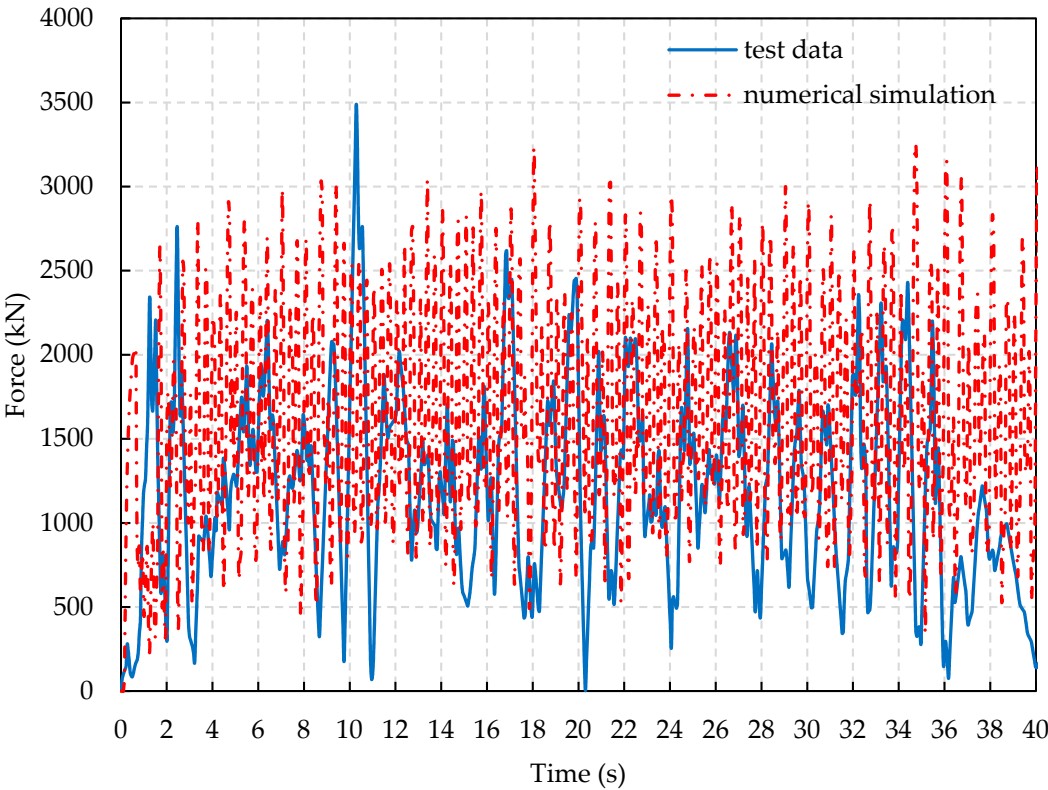

**Figure A1.** Ice force histories from the simulation with mesh size of 0.2 m and measurement for test #304.

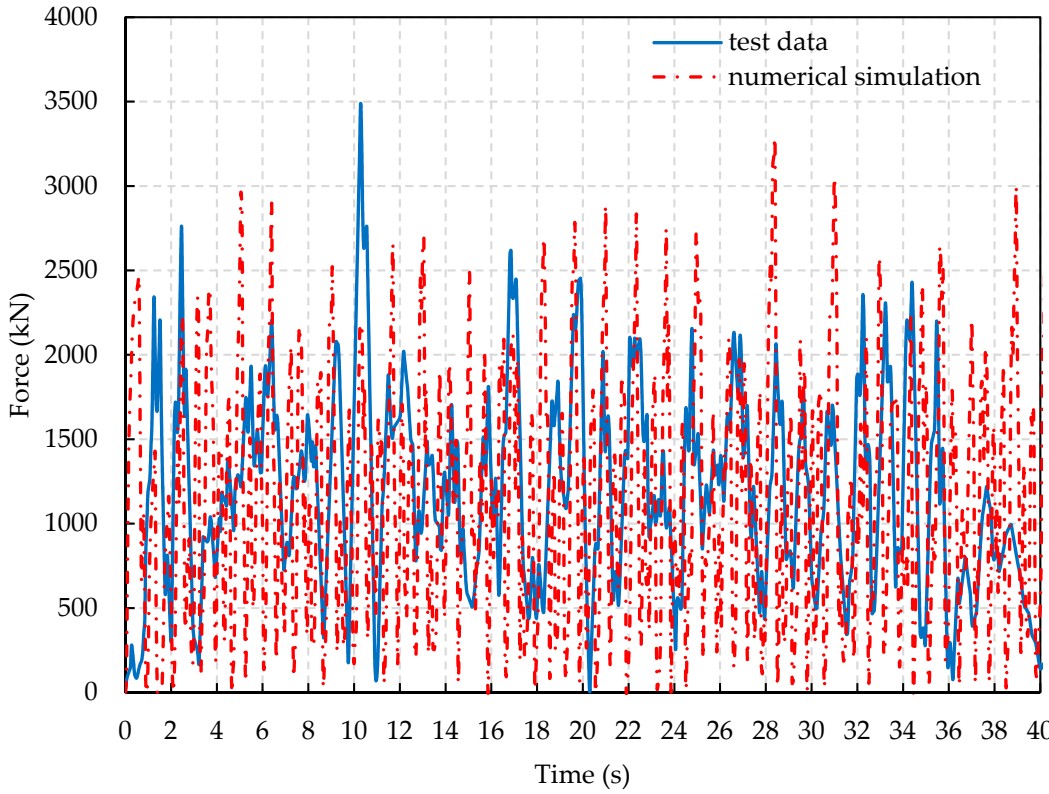

**Figure A2.** Ice force histories from the simulation with mesh size of 0.4 m and measurement for test #304.

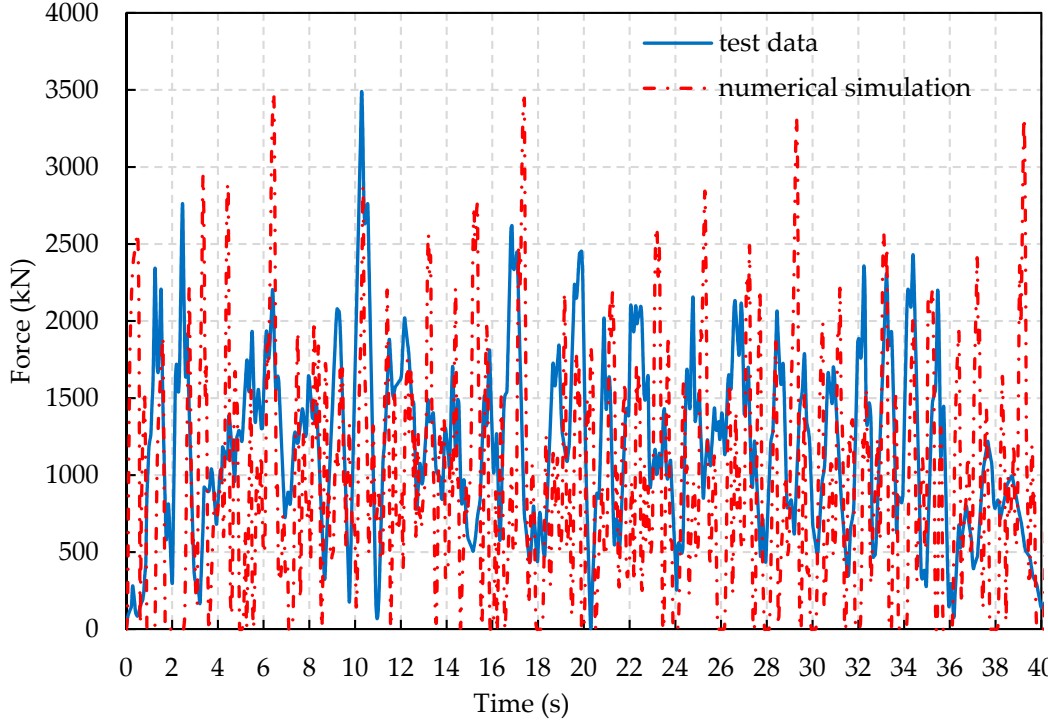

**Figure A3.** Ice force histories from the simulation with mesh size of 0.6 m and measurement for test #304.

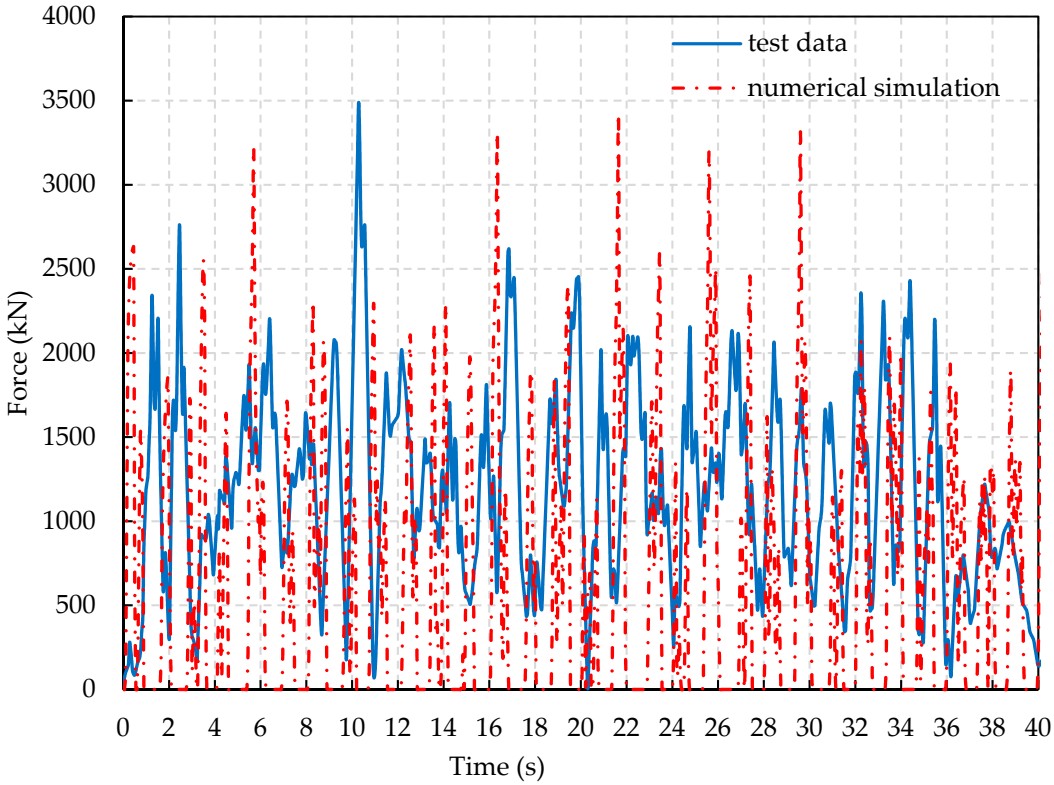

**Figure A4.** Ice force histories from the simulation with mesh size of 0.8 m and measurement for test #304.

## Appendix B. The comparison of the maximum force, mean force, and standard deviation between the simulated and measured results for all tests

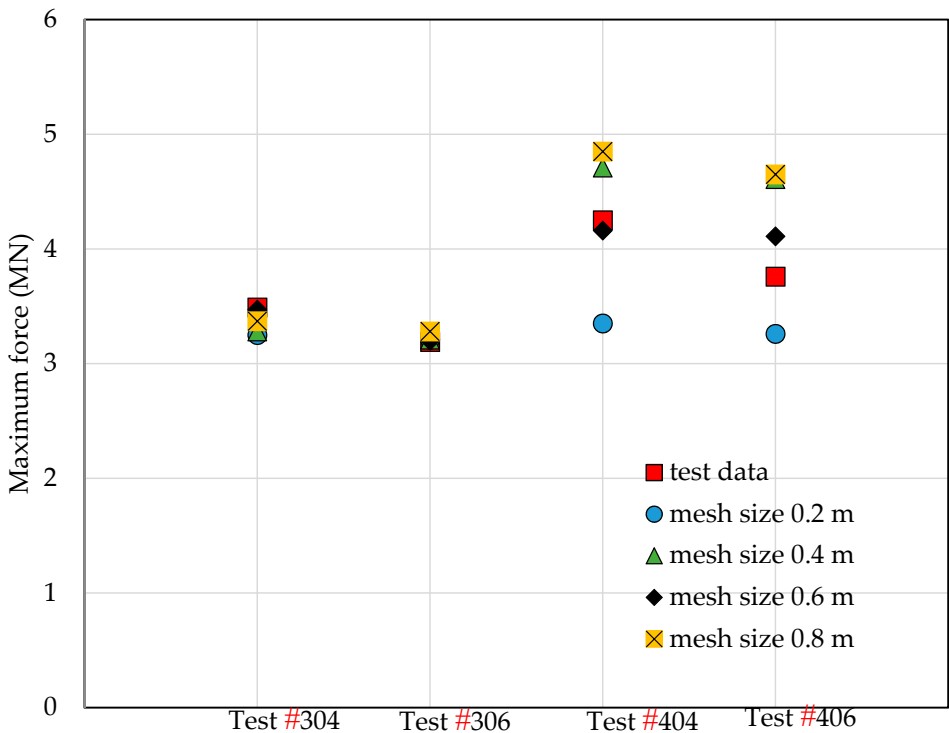



**Figure B1.** Comparison of the maximum ice force between the simulated and measured results for all tests.

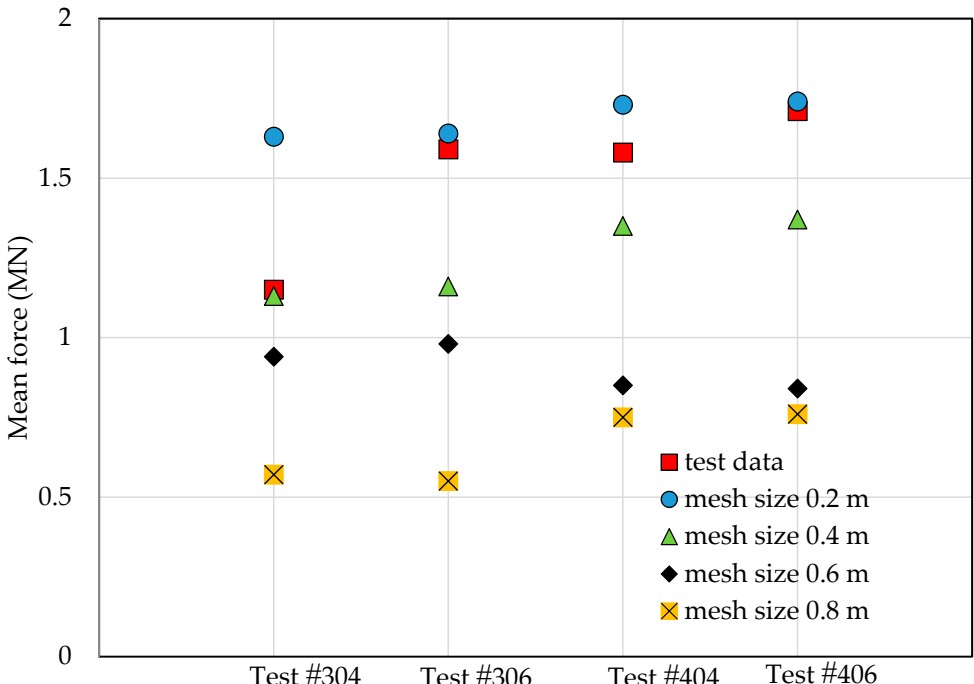

**Figure B2.** Comparison of the mean ice force between the simulated and measured results for all tests.

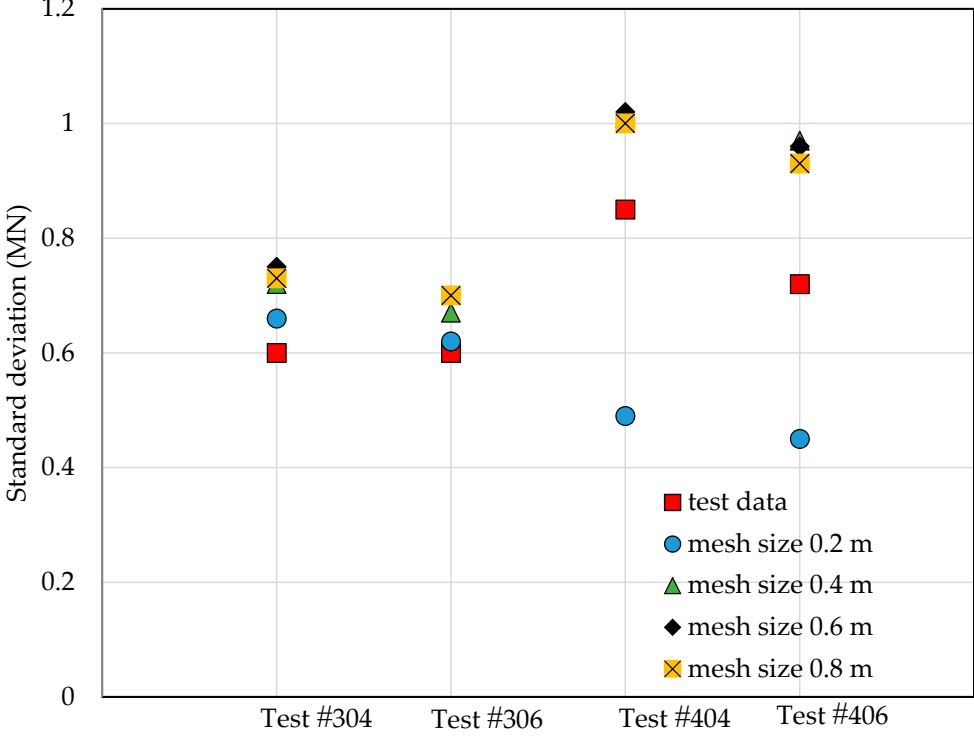

**Figure B3.** Comparison of the standard deviation between the simulated and measured results for all tests.

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
