# Peer review of "Numerical Study of the Interaction between Level Ice and Wind Turbine Tower for Estimation of Ice Crushing Loads on Structure"

_jmse, doi:10.3390/jmse7120439_

Round 1
Reviewer 1 Report
Reviewer’s comments on “Numerical study of the interaction between level ice and wind turbine for estimation of ice crushing loads on structure”
The paper proposed a simulation model using the LS-DYNA code to analyse the interaction between level ice and wind turbine. The below comments are provided to authors to improve their work:
[1] Abstract is long. It can be shortened. Also delete the last sentence “Discussions and conclusions are presented” as this is not needed.
[2] one or two more keywords can be added; such as simulation.
[3] A brief introduction about ice conditions in offshore wind farms will be useful.
[4] It’s not clear why authors used LS-DYNA. Even though this is very useful package available on the market, the authors should list some others and define their pros and cons before they choose LS-DYNA.
[5] The authors review comprehensively some previous research studies investigating the interaction between ice and OWTs, however they didn’t outline the main different between their work and the existing works.
[6] It’s not clear what contribution Section 2 makes to the paper. It seems the experimental data were collected from a reference. So, the reviewer wonders if the authors did also an experiment.
[7] some figures in Section 4 can be transferred to appendix.
Reviewer 2 Report
See the attached file

Round 2
Reviewer 2 Report
According to the last correction, I find the text quite good and extensively improved from the previous version.
I miss (still) the definition of the effective plastic strain. I mean, as the answer reports, it is dimensionless, but it could be clearer for a potential reader, to have its definition.
I would suggest a final reading and minor English-grammar correction of the text.
Author Response
Manuscript ID: jmse-623832
Title: Numerical study of the interaction between level ice and wind turbine for estimation of ice crushing loads on structure
Dear Editor and Reviewers,
Thank you for reviewing our manuscript. We appreciate your valuable comments earnestly, which are helpful to improve the quality of our present study.
According to the comments, we have studied comments carefully point by point and have made correction which we hope to meet with approval.
Revised portion are marked in red in the revised manuscript. The main corrections in the paper and the responds to the reviewer’s comments are as flowing:
Reviewer 2
According to the last correction, I find the text quite good and extensively improved from the previous version.
Response: Authors would like to thank you very much for your precious suggestions and great effort to reviewing the manuscript.
I miss (still) the definition of the effective plastic strain. I mean, as the answer reports, it is dimensionless, but it could be clearer for a potential reader, to have its definition.
Response: Yes. Thank you for this good suggestion. The definition of the effective plastic strain was added in the revision.
I would suggest a final reading and minor English-grammar correction of the text.
Response: Yes. We have checked and corrected the English-grammar in the revision.